# LESS IS MORE: DENOISING KNOWLEDGE GRAPHS FOR RETRIEVAL AUGMENTED GENERATION

## ABSTRACT

Retrieval-Augmented Generation (RAG) systems enable large language models (LLMs) instant access to relevant information for the generative process, demonstrating their superior performance in addressing common LLM challenges such as hallucination, factual inaccuracy, and the knowledge cutoff. Graph-based RAG further extends this paradigm by incorporating knowledge graphs (KGs) to leverage rich, structured connections for more precise and inferential responses. A critical challenge, however, is that most Graph-based RAG systems rely on LLMs for automated KG construction, often yielding noisy KGs with redundant entities and unreliable relationships. This noise degrades retrieval and generation performance while also increasing computational cost. Crucially, current research does not comprehensively address the denoising problem for LLM-generated KGs. In this paper, we introduce DEnoised knowledge Graphs for Retrieval Augmented Generation (DEG-RAG), a framework that addresses these challenges through: (1) entity resolution, which eliminates redundant entities, and (2) triple reflection, which removes erroneous relations. Together, these techniques yield more compact, higher-quality KGs that significantly outperform their unprocessed counterparts. Beyond the methods, we conduct a systematic evaluation of entity resolution for LLM-generated KGs, examining different blocking strategies, embedding choices, similarity metrics, and entity merging techniques. To the best of our knowledge, this is the first comprehensive exploration of entity resolution in LLM-generated KGs. Our experiments demonstrate that this straightforward approach not only drastically reduces graph size but also consistently improves question answering performance across diverse popular Graph-based RAG variants.

## 1 INTRODUCTION

Large Language Models (LLMs) have made significant progress in natural language processing and understanding (Zhao et al., 2023). However, their capabilities are limited by access to up-to-date information, susceptibility to hallucination, and weak long-term memory (Zhao et al., 2023; Huang et al., 2025; Wang et al., 2023). To mitigate these issues, Retrieval-Augmented Generation (RAG) (Lewis et al., 2020) has emerged to ground LLMs with external knowledge. Given a user query, a RAG system retrieves relevant information from a knowledge base, augments the query with the retrieved context, and then generates a response. RAG enables LLMs to access updated information, ground facts, and rapidly adapt to new domain knowledge.

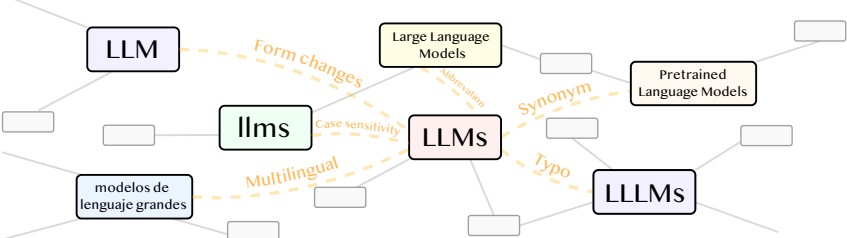

Figure 1: Redundant concept synonyms for "LLMs" in a knowledge graph. Orange dashed lines indicate synonymic equivalences showing why these entities convey the same meaning as "LLMs".

Traditional RAG systems (Karpukhin et al., 2020) retrieve isolated text chunks and ignore relationships among them, which weakens multi-hop reasoning (Yang et al., 2018) and overall coherence (Siriwardhana et al., 2023). Graph-based RAG (Edge et al., 2024; Guo et al., 2024; Jimenez Gutierrez et al., 2024) addresses it by structuring knowledge as a graph and retrieving over that structure. Connectivity among entities allows models to consider inter-document relations rather than treating units as independent chunks, enabling fine-grained, relation-aware retrieval (Hong et al., 2025).

As we all know, the quality of graph is critical to the success of graph mining (Xue & Zou, 2022; Luan et al., 2024), and many graph-based RAG systems focus on constructing knowledge graphs (KGs) from corpora with LLMs. However, the resulting LLM-generated graphs are often noisy and redundant (Huang et al., 2024a). During entity and relation extraction, unlike human experts who can recall and connect new concepts to previously identified entities, LLMs often struggle to consistently maintain earlier entities and relations due to limited long-context capabilities, which leads to duplicates (Lairgi et al., 2024). As illustrated in Figure 1, the extracted entity "LLMs" may co-occur with its variants that represent the same concept, *e.g.,* "LLM" (morphology), "llms" (casing), "modelos de lenguaje grandes" (multilingual), and "Large Language Models" (abbreviation expansion). Existing methods, including LightRAG (Guo et al., 2024), MS GraphRAG [1] (Edge et al., 2024), and HippoRAG (Jimenez Gutierrez et al., 2024), typically rely on string-matching heuristics to merge similar entities, leaving many duplicates unresolved. These **redundant entities** inflate storage, degrade retrieval efficiency and precision. Besides, some outdated and incorrect facts in external corpora(Rietveld et al., 2004; Feng et al., 2025; Moëll & Sand Aronsson, 2025) can yield **erroneous triples** in LLM-generated graphs, which will mislead retrieval and harm generation quality.

To simultaneously reduce the size and improve the quality of generated graphs, we propose DEnoised knowledge Graphs for Retrieval Augmented Generation (DEG-RAG), which takes entity resolution to remove redundancy, and triple reflection to filter erroneous relations in LLM-generated knowledge graphs for RAG. Entity resolution identifies and links records that refer to the same entity (Ebraheem et al., 2017) and is widely used in KG consolidation (Berrendorf et al., 2020). We conduct a comprehensive evaluation and study tailored to Graph-based RAG, spanning different blocking, entity-embedding, matching, and merging strategies.

Our experiments show that, while removing 40% of the entities and relations in LLM-generated KGs, DEG-RAG consistently improves the performance of four representative Graph-based RAG approaches, underscoring the importance of KG quality rather than KG size. We further study the design of different components comprehensively, for example, type-aware blocking is most effective, traditional KG embeddings can rival LLM embeddings, neighborhood-based similarity sometimes outperform ego-based measures, and simple merging often surpasses synonym-edge addition. Together, these findings offer practical guidance for constructing high-quality LLM-generated KGs and for developing more efficient and accurate Graph-based RAG systems, with potential extensions to a wide range of KG-based LLM applications (Choudhary & Reddy, 2023; Wang et al., 2025; Wang, 2025). In summary, our contributions are as follows:

- We propose DEG-RAG, which leverages entity resolution and triple reflection to reduce graph size while improving KG quality for better Graph-based RAG.

- To the best of our knowledge, we are the first to conduct a comprehensive study of entity resolution for Graph-based RAG, implementing and evaluating different components, including blocking, entity-embedding, matching, and merging strategies.

- Our experiments demonstrate that DEG-RAG improves the performance of four graph-based RAG methods across four benchmark QA datasets by removing approximately 40% of entities and relations. We further analyze how different components of entity resolution contribute to Graph-based RAG performance.

## 2 RELATED WORK

Retrieval Augmented Generation (RAG) enables Large Language Models (LLMs) to utilize updated information (Su et al., 2024), access domain-specific knowledge (Zhang et al., 2024), and reduce

---

[1]To avoid ambiguity, we use MS GraphRAG to refer to the specific GraphRAG method proposed in (Edge et al., 2024), and Graph-based RAG to refer to the general class of approaches that leverage knowledge graphs.

hallucinations (Huang et al., 2025). Traditional RAG systems (Karpukhin et al., 2020) organize external knowledge as isolated database chunks, which limits performance in complex reasoning (Yang et al., 2018; Jiang et al., 2024) and contextual completeness (Lu et al., 2025; Zhong et al., 2025). To address these limitations, Graph-based RAG presents external information as graphs, retrieving relevant data by considering inter-relationships (Peng et al., 2024). MS GraphRAG (Edge et al., 2024) constructs communities and generates answers based on community summaries, while LightRAG (Guo et al., 2024) retrieves relevant entities, relationships, and subgraphs using keywords from queries. HippoRAG (Jimenez Gutierrez et al., 2024) employs PageRank (Page et al., 1998) for efficient entity retrieval. KAG (Liang et al., 2024) integrates knowledge graphs (KGs) with LLMs through logical-form-guided reasoning, knowledge alignment, and fine-tuning. Despite these advancements, the quality of LLM-generated KGs remains a challenge, as they are often redundant and noisy, hindering efficient knowledge storage and high-quality generation (Zhou et al., 2025).

Entity resolution, which links data records referring to the same real-world entity, is crucial for constructing high-quality KGs (Pujara & Getoor, 2016; Obraczka et al., 2021). Existing approaches fall into three categories: (1) Traditional methods use string similarity (Yu et al., 2016; Papadakis et al., 2023), heuristic rules (Abu Ahmad & Wang, 2018; Lee et al., 2013), or manually designed schemas (Efthymiou et al., 2019) to identify equivalent entities. These methods are computationally efficient and interpretable but struggle with noisy, incomplete, or multilingual data. (2) Embedding-based methods represent entities in continuous vector spaces, matching based on representation similarity. This includes LLM-based embeddings (Li et al., 2020) and KG embeddings like TransE (Bordes et al., 2013), DistMult (Yang et al., 2014), and ComplEx (Trouillon et al., 2016), as well as Graph Neural Networks (GNNs)-based approaches (Schlichtkrull et al., 2018). These techniques capture structural dependencies across graphs, offering robustness over heuristic methods. (3) LLM-based methods leverage LLMs through prompting (Peeters et al., 2023) or fine-tuning (Steiner et al., 2025) to identify semantically equivalent entities, providing strong generalization capabilities, though they require careful design for scalability and reliability.

Although many entity resolution methods exist, few focus on improving LLM-generated KG quality. For example, MS GraphRAG (Edge et al., 2024) and LightRAG (Guo et al., 2024) use simple string matching for duplicate entity identification. HippoRAG (Jimenez Gutierrez et al., 2024) introduces synonym relations based on cosine similarity, and KAG (Liang et al., 2024) predicts synonym relations from one-hop neighbors, merging entities accordingly. However, the impact of enhancing KG quality on Graph-based RAG is largely unexplored. This paper systematically investigates how different entity resolution methods affect the performance of Graph-based RAG, alongside triple reflection, contributing uniquely beyond previous studies.

## 3 PRELIMINARIES

In this section, we introduce the notations and the process of Graph-based RAG. Given a set of external documents $\mathcal{D} = [d_1, d_2, \ldots, d_N]$, Graph-based RAG constructs a knowledge graph (KG) $\mathcal{G} = (\mathcal{E}, \mathcal{R}, \mathcal{T}, \mathcal{A})$, where $\mathcal{E}$, $\mathcal{R}$, and $\mathcal{T}$ denote the sets of entities, relation types and triples, and $\mathcal{A}$ represents the textual description for each entity. The neighbors of an entity $e \in \mathcal{E}$ are defined as the set of entities $\mathcal{N}(e)$ that are directly connected to $e$ through relation $r \in \mathcal{R}$:

$$\mathcal{N}(e) = \{ e' \in \mathcal{E} \mid (e, r, e') \in \mathcal{T} \ \lor \ (e', r, e) \in \mathcal{T}, \ r \in \mathcal{R} \}. \tag{1}$$

Then, given a user query $Q$, the RAG system (1) retrieves relevant contents from $\mathcal{G}$ via a retrieval function $\mathcal{R}(\cdot)$, (2) augments the query $Q$ with retrieved context using an augmentation function $\text{Aug}(\cdot)$, and (3) generates the final answer $\mathcal{Y}$ with LLMs $\mathcal{M}$. Formally:

$$\mathcal{Y} = \mathcal{M} \circ \text{Aug}\big[Q, \mathcal{R}(Q, \mathcal{G})\big]. \tag{2}$$

Specifically, the raw documents $\mathcal{D}$ are first segmented into text chunks $\mathcal{C} = [c_1, c_2, \ldots, c_M]$. For each chunk $c_m \in \mathcal{C}$, a LLM-based named-entity recognition function $\mathcal{M}_{\text{NER}}(\cdot)$ is applied, leads to a set of raw triples, entities, and relations:

$$\mathcal{T}_m = \mathcal{M}_{\text{NER}}(c_m), \ \mathcal{T} = \bigcup_{m=1}^{M} \mathcal{T}_m, \ \mathcal{E} = \{e_1, e_2 \mid (e_1, r, e_2) \in \mathcal{T}\}, \ \mathcal{R} = \{r \mid (e_1, r, e_2) \in \mathcal{T}\}. \tag{3}$$

where each entity $e \in \mathcal{E}$ carries its local textual context $\mathcal{A}(e)$. Here, the LLM extracted $\mathcal{E}$ may contain duplicates, aliases, or simple variations. To construct a coherent KG, a deduplication function $\phi : \mathcal{E} \mapsto \mathcal{E}^*$ is applied, which maps each raw entity to a unique canonical entity $\phi(e)$. Then we have the revised entity, triple, and relation sets as:

$$\mathcal{E}^* = \{\phi(e) \mid e \in \mathcal{E}\}, \ \mathcal{T}^* = \{(e_1, r, e_2) \mid (e_1, r, e_2) \in \mathcal{T}, e_1 \in \mathcal{E}^*, e_2 \in \mathcal{E}^*\}, \ \mathcal{R}^* = \{r \mid (e_1, r, e_2) \in \mathcal{T}^*\} \tag{4}$$

For each canonical entity $e^* \in \mathcal{E}^*$, we aggregate the textual description with a merge operator $\oplus$:

$$\mathcal{A}^*(e^*) = \bigoplus_{\{e_i : \phi(e_i) = e^*\}} \mathcal{A}(e_i) \tag{5}$$

The final denoised KG is $\mathcal{G}^* = (\mathcal{E}^*, \mathcal{R}^*, \mathcal{T}^*, \mathcal{A}^*)$, enabling more efficient retrieval.

## 4 DENOISING KNOWLEDGE GRAPHS

In most popular Graph-based RAG systems, such as LightRAG (Guo et al., 2024) and MS GraphRAG (Edge et al., 2024), a simple string matching strategy is used as the deduplication function to denoise KGs. However, in this way, entities with the same semantic meaning but different forms, *e.g.,*case sensitivity, abbreviation, synonym, multilingual, and typos, will be missed and isolated from each other. This will lead to a coarse and redundant KG that impedes efficient storage and retrieval in Graph-based RAG systems. To enhance the performance of Graph-based RAG by denoising LLM-generated KGs, we propose to remove redundant entities by entity resolution in Section 4.1 and remove unreasonable edges by triple reflection in Section 4.2. This framework enhances the quality of the KGs while reducing their size.

### 4.1 ENTITY RESOLUTION

Entity resolution for KGs involves several key steps(Christophides et al., 2020), (1) **Blocking:** partitions raw entities into blocks to minimize the number of entity pairs that need to be compared. (2) **Matching and Grouping:** identify entities that represent the same real-world object and then put these matched entities into groups representing a single resolved entity. (3) **Merging and Linking:** combine the raw entities in each cluster into a canonical representation and update the KG by creating or deleting relations as needed. With the above steps, we introduce how to use entity resolution to improve the quality of LLM-generated KGs as follows.

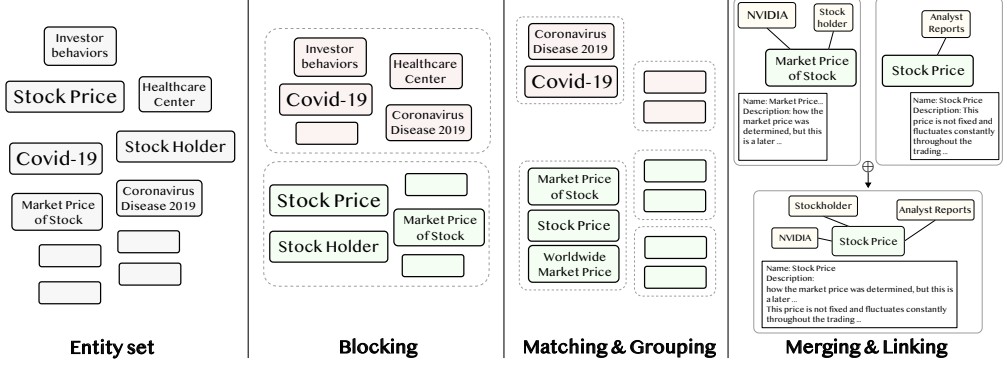

Figure 2: The overall framework of entity resolution for knowledge graphs (Christophides et al., 2020).

**Blocking.** To reduce computational costs and unnecessary entity comparisons, blocking is applied to the entity set $\mathcal{E}$ before entity matching (Papadakis et al., 2019). Formally, blocking is a mapping

$$\text{Block} : \mathcal{E} \mapsto \mathcal{B} = \{B_1, B_2, \ldots, B_K\}, \bigcup_{k=1}^{K} B_k = \mathcal{E} \tag{6}$$

where each block $B_k$ is a subset of entities that are more likely to be matched. In this paper, we consider three types of blocking strategies: semantic-based, entity type-based, and structural-based (Christophides et al., 2020).

(1) Semantic-Based Blocking. Entities are represented as embeddings generated from their descriptions $\mathcal{A}(e)$ using an embedding model $f_{\text{emb}}(\cdot)$. The entity set is partitioned into $k$ clusters by:

$$\mathcal{B} = \texttt{kmeans}\big(\{f_{\text{emb}}(\mathcal{A}(e)) \mid e \in \mathcal{E}\}, k\big),$$

To avoid manual selection of cluster number $k$, we use a rule-of-thumb heuristic $k = \sqrt{\frac{|\mathcal{E}|}{10}}$ (Yuan & Raubal, 2012). This strategy leverages global semantic similarity but is computationally more expensive for large graphs.

(2) Entity Type-Based Blocking. Entities are first classified into types using a type mapping function $\tau : \mathcal{E} \mapsto \Omega$. Entities with the same type $t \in \Omega$ are grouped into the same block:

$$\mathcal{B} = \{\{e \in \mathcal{E} \mid \tau(e) = t\} \mid t \in \Omega\}.$$

If a block contains too many entities, we further subdivide it using $k$-means. The entity type-based blocking limits the matches within the same type of entities, which avoids excessive pair comparisons.

(3) Structural-based Blocking. This strategy exploits graph connectivity under the assumption that semantically similar entities are likely to share neighbors. If an entity $e$ has at least two neighbors, we construct a block for its neighbor set $\mathcal{N}(e)$, and the set of final structural-based blocks is then

$$\mathcal{B} = \{\mathcal{N}(e) \mid e \in \mathcal{E}, |\mathcal{N}(e)| \geq 2\}$$

This blocking is based on the assumption that entities co-occur as neighbors of the same nodes are more likely to present the same meaning, *e.g.,* "Large Language Models" and "Pretrained Language Models" may be placed in the same block if they both connect to the entity "GPU" through the relation "run on.". Therefore, the structural context of shared neighbors serves as a strong signal for blocking.

**Matching and Grouping.** After blocking, the objective is to identify sets of entities in each block that represent the same concept then group entities with the same meaning. Given a block $B \subseteq \mathcal{E}$, the matching function derives a partition:

$$\text{Match} : B \mapsto G = \{G_1, G_2, \ldots, G_L\}, \quad \bigcup_{l=1}^{L} G_l \subseteq B, \tag{7}$$

where each $G_l$ is a group of equivalent entities. To match entities, we first obtain the embedding of each entity $h(e)$ in the KG, then select the entity embedding for matching. Specifically, embedding methods used in this paper include KG embeddings: TransE (Bordes et al., 2013), DistMult (Yang et al., 2014), and ComplEx (Trouillon et al., 2016); graph neural network embeddings: CompGCN (Vashishth et al., 2019) and R-GCN (Schlichtkrull et al., 2018); and LLM embeddings of Qwen3-Embedding-8B (Zhang et al., 2025).

To match similar nodes with proper information after embedding, we consider the calculation of the following similarity scores: (1) **Ego node similarity.** It compares entity embeddings $h(e_i)$ and $h(e_j)$, which is computationally efficient but may miss structural context. (2) **Neighbor similarity.** It compares averaged neighbor embeddings $\bar{h}_{\mathcal{N}}(e_i)$ and $\bar{h}_{\mathcal{N}}(e_j)$, leveraging structural context to identify entities with similar roles. (3) **Type-aware Neighbor similarity.** It compares type-specific averaged neighbor embeddings $\bar{h}_{\mathcal{N}_t}(e_i)$ and $\bar{h}_{\mathcal{N}_t}(e_j)$ for each type $t \in \Omega$, where $\mathcal{N}_t(e) = \{e' \in \mathcal{N}(e) \mid \tau(e') = t\}$, then averages across types: $\text{sim}(e_i, e_j) = \frac{1}{|\Omega|} \sum_{t \in \Omega} \text{sim}_t(\bar{h}_{\mathcal{N}_t}(e_i), \bar{h}_{\mathcal{N}_t}(e_j))$. This reduces noises from irrelevant neighbors and enables precise matching within specific entity types, particularly when entities of different types may have fundamentally different embedding distributions. (4) **Ego+neighbor similarity.** It considers both the ego node and neighbor information by concatenating the embeddings in (1) and (2). (5) **Ego+Type-aware neighbor similarity.** It considers both the ego node and subset of neighbor information by concatenating the embeddings used in (1) and (3). Each matching method captures different aspects of entity similarity and presents distinct trade-offs.

After matching, entities $e_i$ and $e_j$ are grouped together if their similarity exceeds threshold $\delta_{\text{ER}}$, and we assign each entity to a group using the function $g : \mathcal{E} \mapsto G$.

**Merging or Linking.** Once entity groups $G$ are obtained, we finalize the KG $\mathcal{G}^*$ by editing the previous KGs with the following three strategies:

(1) Direct Merging. This approach first selects a single canonical entity $e_l^* = \phi(G_l)$ given a group $G_l$, where $\phi(\cdot)$ refers to a canonical selection function. In this paper, we use random selection for $\phi(\cdot)$. Then, all the other entities inside the group $G_l$ are merged into the canonical entity $\hat{e}_l$.

The KG is updated by appending the descriptions of merged entities to that of the canonical entity, reconnecting their relations to the canonical entity, and removing relations that involve the merged entities. The above process can be expressed as:

$$\mathcal{E}^* = \{\phi(G_l) \mid G_l \in G\}, \ \mathcal{A}^*(\phi(G_l)) = \bigcup_{e \in G_l} \mathcal{A}(e), \ \forall \ G_l \in G \tag{8}$$

$$\mathcal{T}^* = \{\phi(g(e_1)), r, \phi(g(e_2)) \mid (e_1, r, e_2) \in \mathcal{T}, \phi(g(e_1)) \neq \phi(g(e_2))\}. \tag{9}$$

If the merged description of a canonical entity becomes too long, we summarize it to prevent overly long inputs from a single entity during retrieval. The merge of similar entities effectively reduces the storage cost. However, because numerous modifications are made to the original entity and relation sets, the quality of the resulting knowledge graph largely depends on the effectiveness of the entity embedding or matching methods used.

(2) Synonym Linking Only. This approach add a synonym relation $r_{\text{syn}}$ between merged entity $e'$ and canonical entity $\phi(G_l)$ inside each group $G_l$ without the modification of entity set and attributes, which can be described as:

$$\mathcal{T}^* = \mathcal{T} \ \cup \ \{(e', r_{\text{syn}}, \phi(G_l)) \mid e' \in G_l \setminus \phi(G_l), \ G_l \in G_{\text{ent}}\}. \tag{10}$$

This method keeps the minimal changes to the original KG $\mathcal{G}$, yet still cannot well resolve duplication of conceptually similar entities inside $\mathcal{G}$, leading to redundancy and low-efficiency during retrieval.

(3) Merging with Synonym Linking. To prevent the information loss of merged entities as in directly merging, inside each group $G_l$, this approach merges attributes and relations to the canonical entity $\phi(G_l)$ first, then adds synonym relations $r_{\text{syn}}$ towards canonical entity $\phi(G_l)$. In this case, the entity set $\mathcal{E}$ remains unchanged, the relation set $\mathcal{R}$ is updated by Equation (9), then Equation (10), and the attributes is updated by Equation (8).

## 4.2 TRIPLE REFLECTION

Since the external information in the documents may contain erroneous content, the triples extracted by LLMs are not always trustworthy (Huang et al., 2024b; Han et al., 2023). Besides, due to the batched generation of name-entity recognition of chunks, errors may also occur (Lu et al., 2024). Therefore, we use LLM-as-judge to remove the low-quality triple. Specifically, given a triple, composed of source entity, relation, and target entity, we let LLM to predict a reliability score $s = \mathcal{M}_{\text{judge}}(e_1, r, e_2)$. Then, we filter out the triples that are below a threshold $\delta_{\text{TR}}$ and the final relation set that we obtain is

$$\mathcal{T}^* = \{(e_1, r, e_2) \mid (e_1, r, e_2) \in \mathcal{T}, \mathcal{M}_{\text{judge}}(e_1, r, e_2) \geq \delta_{\text{TR}}\} \tag{11}$$

## 4.3 ANALYSIS

Under the construction of KGs in Section 3, if no entity resolution is applied, *i.e.,*the deduplication function becomes identity function, yielding a union of subgraphs with no cross edges. Retrieval over such a disconnected graph reduces to selecting the information of independent triples that a vanilla retriever would select. Formally, we summarize the claim in Proposition 1 as below, where the proof is provided in Appendix D.

**Proposition 1.** Given a graph-based RAG and a vanilla RAG system that share the same augmentation and generation processes, the absence of entity resolution causes the graph-based RAG to degrade into vanilla RAG.

Proposition 1 demonstrates that any benefit of Graph-based RAG over vanilla RAG necessarily comes from the connectivity created by entity resolution.

## 5 EXPERIMENTS

In this section, we comprehensively evaluate the effectiveness the denoising approach mentioned in the previous section for Graph-based RAG systems. We first introduce the experimental settings in Section 5.1. Then, we demonstrate that entity resolution can significantly reduce the scale of the original graph while improving question-answering performance on Graph-based RAG systems in Section 5.2. In Section 5.3, we test and analyze how different components in entity resolution

influence the overall performance. we study the impact of entity reduction ratio and relation reduction ratio on the performance of Graph-based RAG in Section 5.4. Then, we conduct an ablation study in Section 5.5 to evaluate the impact of different deletion methods and LLM API. Additional, we conduct a detailed case study in Appendix B.3 to illustrate the qualitative differences between knowledge graphs before and after the denoising process.

## 5.1 EXPERIMENTAL SETUP

**Datasets and metrics** We evaluate the performance of Graph-based RAG on four datasets from UltraDomain benchmark (Qian et al., 2025) following (Guo et al., 2024), including *Agriculture, CS, Legal*, and *Mix. Agriculture, CS,* and *Legal* contains domain-specific knowledge, while *Mix* includes a broad spectrum of disciplines. Please refer to Appendix A for details of data statistics. Different Graph-based RAG systems are tested by question-answering tasks. We use an LLM as a judge to conduct pairwise comparisons between the responses of two methods, where a winning rate greater than 50% indicates that one method outperforms the other, and vice versa. The evaluation considers four dimensions: comprehensiveness, diversity, empowerment, and overall quality. The detailed evaluation process is shown in Appendix C.5.

**Baselines** We select four popular Graph-based RAG methods as our baselines: (1) LightRAG (Guo et al., 2024). (2) HippoRAG (Jimenez Gutierrez et al., 2024). (3) LGraphRAG (Edge et al., 2024). (4) GGraphRAG (Edge et al., 2024).

**Implementation details.** We implement our experiment based on DIGIMON (Zhou et al., 2025), which is a framework that stably implements many variants of Graph-based RAG and provide a fair and unified comparison among these methods. For efficient indexing and retrieval, the entities and relations are stored in vector dataset bases implemented by Llama Index (Liu, 2022). We use open sourced Qwen3-235B-A22B-Instruct-2507 (Team, 2025) for the LLM API calling, which natively supports 256K context. The model is deployed using VLLM (Kwon et al., 2023) on a Linux server with 8 H20 GPUs. We use Qwen3-Embedding-8B (Zhang et al., 2025) as the embedding model during index building and semantic blocking. For the KG embedding, we use pykeen (Ali et al., 2021), which is design for many types of KG embedding. By default, we set the entity reduction ratio as 40% of the total size of the entity set, $\delta_{\text{TR}}$ of triple reflection as 0.2, semantic-based method for blocking, LLM embeddings for entity embedding, ego-based similarity for matching, direct merging in merging step. Please refer to Appendix C for more implementation details.

Table 1: Performance comparison of graph-based RAG methods on original and cleaned knowledge graphs across four datasets. The evaluation is based on winning rates by comparing responses generated from original versus cleaned knowledge graphs.

| Dataset | Dimension | LightRAG | | HippoRAG | | LGraphRAG | | GGraphRAG | |
|---|---|---|---|---|---|---|---|---|---|
| | | Orig. | Clean | Orig. | Clean | Orig. | Clean | Orig. | Clean |
| Agriculture | Comprehensive | 43.60% | **56.40%** | 49.80% | **50.20%** | 48.80% | **51.20%** | 47.79% | **52.21%** |
| | Diversity | 41.60% | **58.40%** | 43.78% | **56.22%** | 40.00% | **60.00%** | 36.14% | **63.86%** |
| | Empowerment | 42.00% | **58.00%** | 47.39% | **52.61%** | 45.60% | **54.40%** | 47.79% | **52.21%** |
| | Overall | 42.40% | **57.60%** | 48.19% | **51.81%** | 47.20% | **52.80%** | 47.39% | **52.61%** |
| CS | Comprehensive | 39.20% | **60.80%** | 49.17% | **50.83%** | 47.18% | **52.82%** | 48.19% | **51.81%** |
| | Diversity | 40.00% | **60.00%** | 35.54% | **64.46%** | 43.55% | **56.45%** | 44.58% | **55.42%** |
| | Empowerment | 40.80% | **59.20%** | 49.17% | **50.83%** | 47.58% | **52.42%** | 48.59% | **51.41%** |
| | Overall | 41.60% | **58.40%** | 49.59% | **50.41%** | 46.77% | **53.23%** | 48.19% | **51.81%** |
| Legal | Comprehensive | 43.60% | **50.80%** | 49.60% | **50.40%** | 44.80% | **55.20%** | 48.00% | **52.00%** |
| | Diversity | 41.60% | **51.20%** | 44.00% | **56.00%** | 36.80% | **63.20%** | 42.80% | **57.20%** |
| | Empowerment | 42.00% | **51.60%** | 50.00% | 50.00% | 45.20% | **54.80%** | 48.00% | **52.00%** |
| | Overall | 42.40% | **51.60%** | 50.00% | 50.00% | 44.80% | **55.20%** | 47.60% | **52.40%** |
| Mix | Comprehensive | 45.60% | **54.40%** | 48.80% | **51.20%** | 45.20% | **54.80%** | 49.60% | **50.40%** |
| | Diversity | 40.80% | **59.20%** | 51.60% | 48.40% | 38.40% | **61.60%** | 45.20% | **54.80%** |
| | Empowerment | 45.60% | **54.40%** | 47.60% | **52.40%** | 42.40% | **57.60%** | 49.20% | **50.80%** |
| | Overall | 46.00% | **54.00%** | 48.40% | **51.60%** | 42.40% | **57.60%** | 49.40% | **50.60%** |

## 5.2 IMPACT OF KNOWLEDGE GRAPH DENOISING

To validate the effectiveness of our proposed DEG-RAG, we compare the performance of baseline Graph-based RAG with denoised KGs and original KGs on four datasets. As shown in Table 5.1,

after reducing $40\%$ of the entities and removing erroneous relations, the performance of Graph-based RAG on cleaned KGs is better than the original KGs in most cases. This indicates the necessity of denoising KGs for Graph-based RAG. Note that for HippoRAG, the performance is not significantly improved on the *Legal* and *Mix* datasets. This is because the entity set of the KG in HippoRAG only contains entity names without descriptions, limiting the performance of entity resolution.

## 5.3 COMPONENT ANALYSIS OF ENTITY RESOLUTION

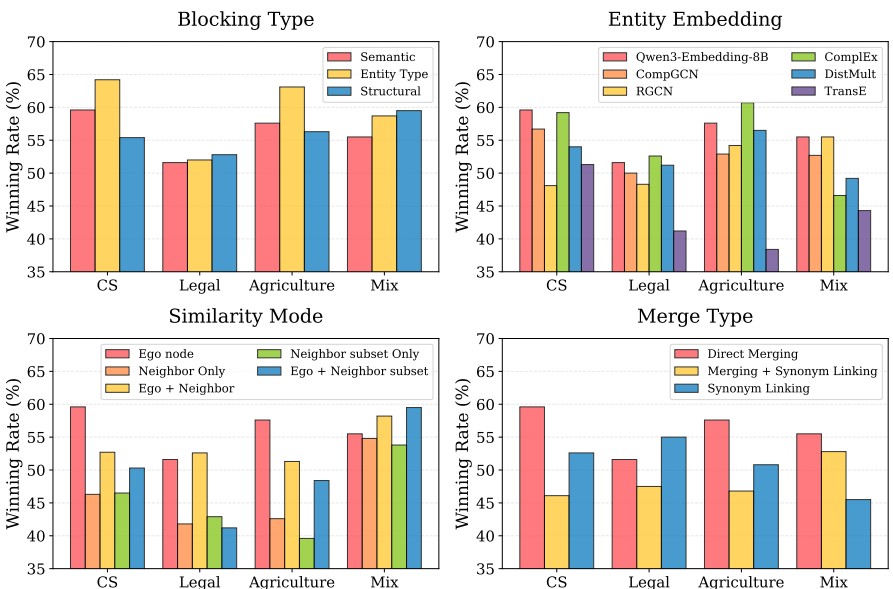

Figure 3: Impact of different entity resolution components on Graph-based RAG performance.

We further study the impact of different components of entity resolution on the performance of Graph-based RAG. Figure 3 shows the winning rate averaged across four metrics (Comprehensive, Diversity, Empowerment, and Overall) on denoised KGs with different components of blocking type, entity embedding, similarity mode, and merge type. We find that: (1) Entity type-based blocking is more effective than semantic-based or structure-based blocking. We speculate that entity type is a better and more natural inductive bias for entity resolution and can lead to more robust denoised graph, which is important for graph mining (Luan et al., 2022; Zheng et al., 2024). (2) Traditional KG embeddings can rival LLM embeddings. In the *Legal* and *Agriculture* datasets, LLM embeddings underperform ComplEx embeddings (Trouillon et al., 2016), which represents entities and relations as vectors in a complex number vector space to better handle asymmetric relations. This demonstrates that traditional KG embeddings can be a viable alternative to LLM embeddings, especially in scenarios where computational resources are insufficient for LLMs or when we contain complex relations in the datasets. (3) Without ego-based similarity, the performance of Graph-based RAG degrades in most cases. Additionally, incorporating neighbor information as a complement to ego node information improves performance in the *Legal* and *Mix* datasets. (4) Simple direct merging often surpasses synonym linking. Although both methods aim to deal with the synonym entities, synonym linking only adds synonym relations between merged entities and the canonical entity. As a result, the KGs remain redundant, requiring more hops to retrieve relevant information. In contrast, direct merging addresses this by consolidating entities with similar meanings into a single entity, which is more efficient.

## 5.4 HYPERPARAMETER ANALYSIS

We conduct experiments to investigate the robustness of the selection of the entity reduction ratio on the effectiveness of denoising. As shown in Figure 4, the winning rate is equal or larger than $50\%$ as long as reduction ratio is not too high. This means, as long as entities are not over-merged, the denoising step is effective for Graph-based RAG. Notably, on *Mix* and *Legal*, the performance

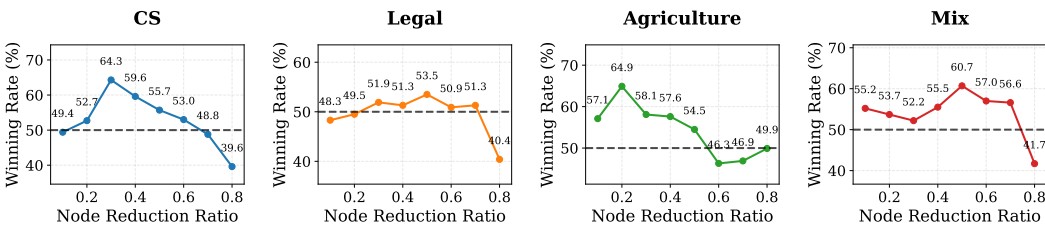

Figure 4: Influence of entity reduction ratio on Graph-based RAG performance.

remains comparable to the original KG up to 70%, which means even the reduction of 70% entities in KG does not cause negative effect compared to original KG. At such aggressive denoising setting, not only near-duplicate or synonymous entities are merged, but entities with only marginal semantic similarity and overlapping local neighborhoods can also be absorbed into a single canonical node, effectively collapsing fine-grained clusters. The resulting KG becomes substantially more compact while still keep, and sometimes even improve, Graph-based RAG performance. We attribute this to the reduced redundancy, shorter multi-hop paths, and the concentration on fewer, more informative nodes. This indicates that Graph-based RAG is robust to some over-merging cases so long as coarse-grained semantics are preserved.

## 5.5 ABLATION STUDY

To evaluate the effectiveness of entity reso-
lution and triple reflection in DEG-RAG, we
conduct an ablation study in this subsection.
As shown in Figure 5, without entity reso-
lution or triple reflection, the performance of
Graph-based RAG significantly degrades in all
datasets. Moreover, we find that entity reso-
lution is more impactful than triple reflection,
indicating the necessity of entity resolution in
KGs. We also set up random merging as a ref-
erence method for comparison and the results
show worse performance than the above two
partial methods, which again shows the neces-
sity to handle the redundant entities smartly.

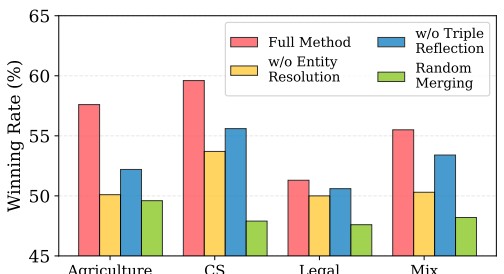

Figure 5: Ablation study on the performance of the full denoising method against versions without entity resolution, without triple reflection, and with random entity merging.

## 6 CONCLUSION AND FUTURE WORKS

In this work, we investigated how denoising LLM-generated KGs benefits Graph-based RAG. We introduced DEG-RAG, which combines entity resolution and triple reflection to remove redundant entities and filter unreliable relations. Across four Graph-based RAG variants and four datasets, DEG-RAG reduces around half the size of the entities and relations while preserving or improving QA quality and lowering storage cost. Our component analysis shows that type-aware blocking is consistently strong, classical KG embeddings such as ComplEx can rival LLM embeddings, ego information is essential and neighbor cues help in some settings, and direct merging generally outperforms synonym-only linking. Hyperparameter sweeps reveal wide operating regimes and sometimes allow up to 70% entity reduction without hurting performance. Our methods focus on improving the quality of KGs and can be used alongside advances in knowledge-graph-based LLM applications (Choudhary & Reddy, 2023; Wang et al., 2025; Wang, 2025).

While effective, DEG-RAG has limitations. Our study uses four QA datasets and non-large-scale KGs. Triple reflection depends on LLM prompting and the LLM-as-judge setup, which can intro-duce calibration bias. Gains are bounded by attribute richness. For example, graphs with only short names without rich descriptions limit resolution quality. In future work, we will extend DEG-RAG to more datasets and larger-scale KGs, generalize the denoising pipeline to other LLM-generated data structures beyond KGs, and richer evaluations beyond LLM as judges.

REPRODUCIBILITY STATEMENT

We have provided the codebase in supplementary material and all the results in this paper are reproducible. The additional implementation details and experimental setups can be found in Section 5.1 and Appendix C.

ETHICS STATEMENT

All of the authors in this paper have read and followed the ethics code.

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

# THE USE OF LARGE LANGUAGE MODELS

In this work, we employed LLMs as auxiliary tools to support the preparation of the manuscript. Specifically, LLMs were used in two ways: (i) to polish the writing style of the paper by refining grammar, clarity, and readability without altering the technical content, and (ii) to assist in identifying relevant related work by suggesting potential references. Note that LLMs were not involved in designing experiments, analyzing results, or drawing conclusions; these aspects of the study were carried out independently by the authors.

## A  DATA STATISTICS

Table 2: Statistics of datasets and knowledge graphs across four domains.

| Category | | Agriculture | CS | Legal | Mix |
|---|---|---|---|---|---|
| # Token | | 1,949,526 | 2,047,866 | 4,872,343 | 611,161 |
| # Document | | 12 | 10 | 94 | 61 |
| # Question | | 125 | 125 | 125 | 125 |
| # Entity | LightRAG | 21,131 | 16,434 | 16,502 | 8,942 |
| | HippoRAG | 42,444 | 25,495 | 34,342 | 24,055 |
| | LGraphRAG | 21,761 | 15,257 | 16,761 | 10,240 |
| | GGraphRAG | 21,227 | 15,600 | 16,111 | 10,399 |
| # Relation | LightRAG | 23,102 | 20,642 | 33,625 | 7,458 |
| | HippoRAG | 41,636 | 25,170 | 51,031 | 16,370 |
| | LGraphRAG | 25,834 | 19,980 | 36,742 | 8,513 |
| | GGraphRAG | 21,408 | 19,412 | 36,507 | 9,943 |
| Ave. Entity Description | LightRAG | 40.47 | 42.12 | 63.64 | 32.61 |
| | HippoRAG | – | – | – | – |
| | LGraphRAG | 40.23 | 40.21 | 62.11 | 31.88 |
| | GGraphRAG | 38.74 | 39.83 | 63.76 | 33.66 |

As shown in Table A, we report the numbers of tokens, documents, and questions for the four datasets used in this paper. We also present the counts of entities and relations, as well as the average length of entity descriptions (in tokens) in the LLM-generated knowledge graphs extracted by LightRAG (Guo et al., 2024), HippoRAG (Jimenez Gutierrez et al., 2024), LGraphRAG (Edge et al., 2024), and GGraphRAG (Edge et al., 2024). Note that the knowledge graphs generated by HippoRAG do not contain entity descriptions.

## B  ADDITIONAL EXPERIMENTAL RESULTS

### B.1  IMPACT OF DIFFERENT LLMS IN RAG

To show how different LLMs backbones influences the performance of DEG-RAGshown in Table 5.1, apart from Qwen3-235B-A22B (Team, 2025), we further conduct experiments using GPT-4o-mini (OpenAI, 2024) and Gemini-2.5-flash (Comanici et al., 2025) on four datasets on LightRAG (Guo et al., 2024). As shown in Table B.1, under the entity reduction of $40\%$ and triple reflection threshold of $0.2$, the winning rate of using GPT-4o-mini or Gemini-2.5-flash is comparable as Qwen3-235-A22B, indicating the generality of DEG-RAGacross different types of LLMs.

### B.2  COMPARISON OF TOKEN CONSUMPTION

We further compare the costs of DEG-RAGunder different entity reduction ratios. Table B.2 shows the statistics of token consumption after applying DEG-RAGin LightRAG as shown in Table 5.1. First, we can see that there is no significant differences of token consumption in prompt and completion for LightRAG on the original knowledge graph and knowledge graphs with DEG-RAG, indicating the performance gain is not caused by additional information. Second, we notice that the

Table 3: Performance comparison of models on original and cleaned knowledge graphs across four datasets. The evaluation is based on winning rates by comparing responses generated from original versus cleaned knowledge graphs.

| Dataset | Dimension | Qwen3-235B-A22B | | GPT-4o-mini | | Gemini-2.5-flash | |
| | | Orig. | Clean | Orig. | Clean | Orig. | Clean |
|---|---|---|---|---|---|---|---|
| Agriculture | Comprehensive | 43.60% | **56.40%** | 45.34% | **54.66%** | 46.00% | **54.00%** |
| | Diversity | 41.60% | **58.40%** | 29.27% | **70.73%** | 46.00% | **54.00%** |
| | Empowerment | 42.00% | **58.00%** | 31.71% | **68.29%** | 46.80% | **53.20%** |
| | Overall | 42.40% | **57.60%** | 33.74% | **66.26%** | 46.80% | **53.20%** |
| CS | Comprehensive | 39.20% | **60.80%** | 42.32% | **57.68%** | 44.40% | **55.60%** |
| | Diversity | 40.00% | **60.00%** | 36.51% | **63.49%** | 43.20% | **55.60%** |
| | Empowerment | 40.80% | **59.20%** | 41.91% | **58.09%** | 43.20% | **56.80%** |
| | Overall | 41.60% | **58.40%** | 41.91% | **58.09%** | 44.00% | **56.00%** |
| Legal | Comprehensive | 43.60% | **56.40%** | 46.40% | **53.60%** | 42.00% | **58.00%** |
| | Diversity | 41.60% | **58.40%** | 45.20% | **54.80%** | 42.40% | **57.60%** |
| | Empowerment | 42.00% | **58.00%** | 46.80% | **53.20%** | 40.80% | **59.20%** |
| | Overall | 42.40% | **57.60%** | 47.60% | **52.40%** | 41.20% | **58.80%** |
| Mix | Comprehensive | 45.60% | **54.40%** | 47.18% | **52.82%** | 42.40% | **57.60%** |
| | Diversity | 40.80% | **59.20%** | 43.95% | **56.05%** | 40.00% | **60.00%** |
| | Empowerment | 45.60% | **54.40%** | 45.16% | **54.84%** | 42.40% | **57.60%** |
| | Overall | 46.00% | **54.00%** | 45.56% | **54.44%** | 42.00% | **58.00%** |

Table 4: Token consumption statistics under different entity reduction ratios across four datasets.

| Dataset | Type | Original | 20% | 40% | 60% | 80% |
|---|---|---|---|---|---|---|
| Mix | Prompt | 1,040,189 | 1,185,787 | 1,267,955 | 1,149,659 | 1,133,338 |
| | Completion | 86,171 | 85,738 | 85,454 | 85,334 | 86,051 |
| | Total Token | 1,126,360 | 1,271,525 | 1,353,409 | 1,234,993 | 1,219,389 |
| CS | Prompt | 1,084,623 | 1,118,326 | 1,106,513 | 906,618 | 779,191 |
| | Completion | 89,056 | 90,658 | 89,394 | 88,844 | 89,252 |
| | Total Token | 1,173,679 | 1,208,984 | 1,195,907 | 995,462 | 868,443 |
| Agriculture | Prompt | 1,273,710 | 1,537,191 | 1,296,947 | 1,278,717 | 911,124 |
| | Completion | 82,351 | 82,677 | 82,978 | 79,724 | 79,683 |
| | Total Token | 1,356,061 | 1,619,868 | 1,379,925 | 1,358,441 | 990,807 |
| Legal | Prompt | 1,755,056 | 1,749,183 | 1,721,740 | 1,528,838 | 1,658,700 |
| | Completion | 84,124 | 84,771 | 84,178 | 83,707 | 85,468 |
| | Total Token | 1,839,180 | 1,833,954 | 1,805,918 | 1,612,545 | 1,744,168 |

input token increases with node reduction of 20% or 40%, then decreases on 60% and 80%. We explain this as, in lower reduction ratio, few entities are merged, which slightly increases the input prompt, while in high reduction ratio, more and more entities are merged together, after the summaziation of entitiy description, the total retrieved entites and relations become fewer, leads to fewer input token.

## B.3 CASE STUDY

To illustrate the qualitative impact of denoising, we conduct a case study on entity resolution using the CS dataset. Figure 6 shows a subgraph of the knowledge graph before and after denoising. Red nodes indicate redundant entities that have been merged into their canonical forms, while blue nodes represent entities that remain unchanged. Dashed red lines indicate the direction of merging from one entity to another, green lines denote newly added relations, brown dashed lines represent removed relations, and black lines correspond to relations that are retained.

The entity merging process is generally reasonable. For example, variations such as `ARIME methodology` are merged into `ARIMA model`, and `Linear Regression` into `linear`

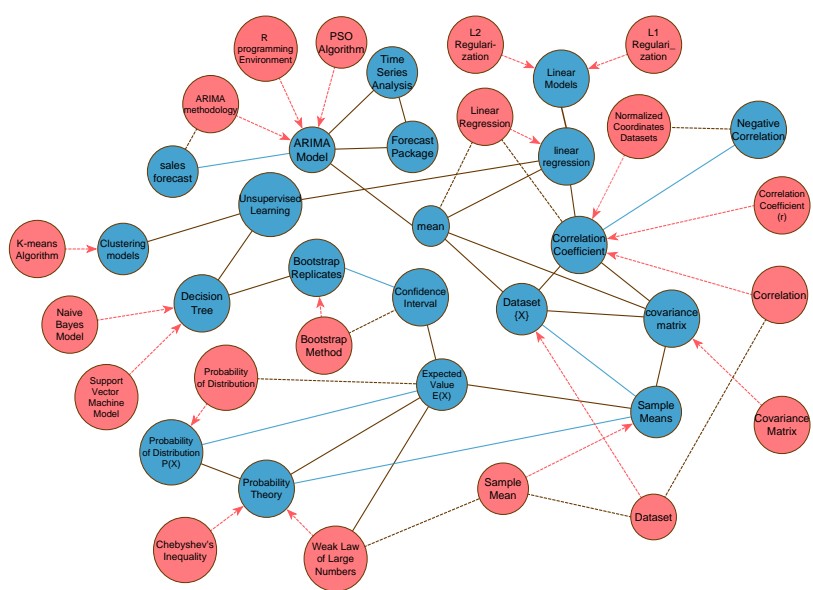

Figure 6: Case Study of Knowledge Graph Denoising on the CS Dataset. The figure illustrates a subgraph before and after applying our denoising method. Redundant entities are denoted in red and merging process is shown in arrows.

regression. We also observe merges driven by semantic similarity, such as `K-means Algorithm` being merged into `Clustering models`, and `Naive Bayes Model` and `Support Vector Machine Model` being merged into `Decision Tree`. Overall, the de-noised knowledge graph is more concise and efficient, thereby improving the performance of graph-based RAG.

We also examine the cases of triple reflection. As in Table 5, we listed some triples with $\delta_{\text{TR}} \leq 0.2$.

## C  IMPLEMENTATION DETAILS

### C.1  GRAPH-BASED RAG

For all the Graph-based RAG methods, we set token-based chunking across all methods, with segment length of approximately 1,200 tokens and an overlap of 100 tokens, using a standard tokenizer to balance context preservation and indexing granularity. We set the retriever to return the top 5 candidates. When personalized PageRank is used, we set entity-aware priors with light damping to encourage focus on salient nodes. All methods answer questions directly rather than only returning supporting context. We set the overall candidate pool to 20. We set token budgets consistently across methods: the naive assembly budget to 12,000 tokens, the local assembly budget to 4,000 tokens, and the entity and relation evidence budgets to 2,000 tokens each. When iterative reasoning over retrieved evidence is enabled, we cap the refinement steps at 2.

LightRAG (Guo et al., 2024) maintains both entity and relation indices and builds a relation-centric knowledge graph enriched with edge keywords. We enable entity descriptions, entity types, edge descriptions, and edge names to maximize semantic coverage. We set the usable context window to 32,768 tokens. For retrieval, we set nearest-neighbor search and enable entity-similarity–aware propagation with the top 5 results. Querying is hybrid: we enable both local and global graph search. We set the global community cap to 512 without a minimum rating, the global community report budget to 16,384 tokens, and the global context budget to 4,000 tokens. Locally, we set the context budget to 4,800 tokens and the community report budget to 3,200 tokens. We allow keyword cues when composing the final context.

Table 5: Case study of triple reflection

| Relation | Source | Score | Target | Analysis |
|---|---|---|---|---|
| Transenterix Inc. owns Safestitch LLC | Transenterix Inc. owns Safestitch LLC, indicating a parent subsidiary relationship | 0.1 | Safestitch LLC | TransEnterix does not own SafeStitch |
| Turtle is one of the entities classified as a borrower | Turtle is one of the entities classified as a borrower in the financial agreement | 0.1 | Borrowers | Turtles are not entities that engage in borrowing |
| Michael Scott is involved in the SEC lawsuit | Michael Scott is involved in the SEC lawsuit as a defendant accused of securities violations | 0.1 | SEC lawsuit | Michael Scott is a fictional character from the television show 'The Office' and not a real person involved in any legal matters |
| Title policy for Pabst | Title policy is required to obtain a title policy to ensure the legitimacy of the asset ownership during the acquisition | 0.1 | Pabst | A title policy is a type of insurance related to real estate transactions, while 'pabst' appears to refer to a brand |
| Shareholder's equity reflects net worth of dealers | Shareholder's equity is a key financial metric that reflects the net worth of dealers after liabilities are deducted | 0.2 | Dealers | Shareholder's equity is a financial metric relevant to companies and their owners, not specifically to dealers |
| Kristen M Jenner and Kylie K Jenner are key executives | Kylie K Jenner and Kristen M Jenner are both identified as key executives, indicating a professional relationship in a business context | 0.2 | Kylie K Jenner | Kristen M Jenner is not a recognized executive in the same context as Kylie K Jenner |

HippoRAG (Jimenez Gutierrez et al., 2024) focuses on an entity–relation graph with entity-link–aware chunking and enables graph augmentation while keeping metadata conservative: we disable entity and edge descriptions, and we retain edge names. We set retrieval to personalized PageRank over the entity–relation graph without an entity-similarity term in propagation, and we set the top-$k$ to 5. Querying follows a hybrid strategy while we disable explicit propagation-based augmentation in the final context assembly. We keep the same token budgets as in the common configuration, and we cap iterative reasoning at 2 steps.

LGraphRAG (Edge et al., 2024) uses a relation-centric knowledge graph with a forced construction setting. We enable entity and edge descriptions and edge names, and we disable entity types. We apply community-aware clustering using the Leiden algorithm; we set the maximum community size to 10 and use concise community summaries. We set retrieval to nearest-neighbor search with an additional local neighborhood expansion, and we enable propagation-based augmentation while disabling global community selection. We set the local context budget to 4,800 tokens and the local

community report budget to 3,200 tokens, and we keep the same overall budgets and refinement limits as in the common setup.

GGraphRAG (Edge et al., 2024) adopts the same relation-centric graph construction and community-aware clustering as LGraphRAG. We set retrieval to nearest-neighbor search without local expansion, and we enable both local and global querying. We set the global community cap to 512, the global community report budget to 16,384 tokens, and the global context budget to 4,000 tokens, while keeping the local budgets aligned with the common configuration. Other token allocations and refinement limits follow the common setup.

## C.2 REDUCTION RATIO

We further report the number and proportion of removed entities and relations in Table 5.1. As shown in Table 6, across the four datasets, the entity reduction ratio is approximately $40\%$. The relation reduction ratio ranges from $30\%$ to $60\%$, reflecting both the removal of relations during triple reflection and the disappearance of relations associated with merged entities.

Table 6: Statistics of original and cleaned knowledge graphs across four datasets and four Graph-based RAG models.

| Dataset | Dimension | LightRAG | | | HippoRAG | | | LGraphRAG | | | GGraphRAG | | |
|---|---|---|---|---|---|---|---|---|---|---|---|---|---|
| | | Orig. | Clean | Reduction | Orig. | Clean | Reduction | Orig. | Clean | Reduction | Orig. | Clean | Reduction |
| Agriculture | # Entity | 21131 | 12679 | 40.00% | 42444 | 25466 | 40.00% | 21761 | 13057 | 40.00% | 21227 | 12736 | 40.00% |
| | # Relation | 23102 | 15548 | 32.70% | 41636 | 20321 | 51.19% | 25834 | 16503 | 36.12% | 21408 | 11258 | 47.41% |
| CS | # Entity | 16434 | 9861 | 40.00% | 25495 | 15297 | 40.00% | 15257 | 9154 | 40.00% | 15600 | 9360 | 40.00% |
| | # Relation | 20642 | 12164 | 41.07% | 25170 | 13801 | 45.17% | 19980 | 13756 | 33.15% | 19412 | 13742 | 29.21% |
| Legal | # Entity | 16502 | 9902 | 40.00% | 34342 | 20606 | 40.00% | 16761 | 10057 | 40.00% | 16111 | 9667 | 40.00% |
| | # Relation | 33625 | 21261 | 36.77% | 51031 | 35920 | 29.61% | 36742 | 22987 | 37.44% | 36507 | 14025 | 61.58% |
| Mix | # Entity | 8942 | 5366 | 40.00% | 24055 | 14433 | 40.00% | 10240 | 6144 | 40.00% | 10399 | 6240 | 40.00% |
| | # Relation | 7458 | 5164 | 30.76% | 16370 | 6896 | 57.87% | 8513 | 6288 | 26.14% | 9943 | 6713 | 32.49% |

## C.3 PROMPTS IN ENTITY RESOLUTION

To avoid the exceeding length of descriptions of merged knowledge graphs, we summarize the descriptions if the number of token exceed 4,000. We provide the summarization prompt of entity and relation as follows

---

**Entity description summarization prompt**

You are a helpful assistant. Please summarize the following list of descriptions for the entity {entity_name} into a single, coherent paragraph. Combine the key information and remove redundant details.
Descriptions to summarize:
{description_list}
**Concise Summary:**

---

**Relation description summarization prompt**

You are a helpful assistant. Please summarize the following list of descriptions for the relationship {item_name} into a single, coherent paragraph. Combine the key information and remove redundant details.
Descriptions to summarize:
{description_list}
**Concise Summary:**

---

## C.4 Prompts in Triple Reflection

We perform triple reflection on knowledge graph triples (edges) using LLMs to assess their reasonableness before downstream use. For each triple, an LLM returns a numerical quality score and a short analysis; results are written as JSONL for subsequent aggregation and filtering.

---

**System prompt**

You are a knowledge graph expert who evaluates whether the knowledge graph triplet belongs to commonsense knowledge.

---

**User prompt**

Evaluate the reasonableness of the knowledge graph triplet with precision:

Source: `<source>`
Destination: `<destination>`
Relationship: `<relationship>`

**Analysis requirements**

- **Semantic accuracy**: Does the relationship accurately describe the connection? Consider domain knowledge and factual correctness.

- **Relevance**: Is the connection meaningful and significant, not trivial or coincidental?

- **Specificity**: Is the relationship clear and specific rather than vague or overly general?

- **Logical coherence**: Does the triple follow expected semantic and syntactic patterns for KGs?

- **Entity type compatibility**: Is the relationship sensible given the entity types involved?

**Scoring guidelines**

- 0.0–0.3: Invalid or highly questionable (factually wrong, illogical, meaningless)

- 0.4–0.6: Partially valid but problematic (some relevance yet vague/imprecise/minor inaccuracies)

- 0.7–0.8: Mostly valid (accurate but could be more specific or informative)

- 0.9–1.0: Fully valid (accurate, specific, informative, and logically sound)

**Optimization notes**

- Focus on direct evaluation without unnecessary elaboration.

- Use domain-specific reasoning where applicable.

Output format (return a valid JSON object):

```
{
    "analysis": "concise analysis",
    "score": 0.5
}
```

The score should be a float between 0.0–1.0 with two-decimal precision.

---

## C.5 Evaluation

We assess the responses of DEG-RAG using an LLM judge in a pairwise-comparison setup. For each question the judge receives the question and two candidate answers from original knowledge graphs or denoised knowledge graphs by DEG-RAG, and decides which answer is better and why. To mitigate position bias we run two passes per question. Pass A uses (Answer 1, Answer 2) and Pass B swaps the order. Aggregated wins for a method on a criterion are computed by summing Answer 1 wins in Pass A and Answer 2 wins in Pass B. Ties are recorded when the judge issues a tie token. The judge receives the following prompts verbatim.

**System prompt**

You are an expert tasked with evaluating two answers to the same question based on three criteria: **Comprehensiveness**, **Diversity**, and **Empowerment**.

**User prompt**

You will evaluate two answers to the same question using the three criteria below:

- **Comprehensiveness**: How much detail does the answer provide to cover all aspects and details of the question?

- **Diversity**: How varied and rich is the answer in presenting different perspectives and insights?

- **Empowerment**: How well does the answer help the reader understand the topic and make informed judgments?

For each criterion, choose the better answer (**Answer 1** or **Answer 2**) and explain why. Then select an overall winner based on these three categories.
Here is the question: {query}
Here are the two answers:
**Answer 1:** {answer1}
**Answer 2:** {answer2}
Evaluate both answers using the three criteria above and provide detailed explanations for each criterion.
Output your evaluation in the following JSON format:

```
{
    "Comprehensiveness": {
        "Winner": "[Answer 1 or Answer 2]",
        "Explanation": "[Provide explanation here]"
    },
    "Diversity": {
        "Winner": "[Answer 1 or Answer 2]",
        "Explanation": "[Provide explanation here]"
    },
    "Empowerment": {
        "Winner": "[Answer 1 or Answer 2]",
        "Explanation": "[Provide explanation here]"
    },
    "Overall Winner": {
        "Winner": "[Answer 1 or Answer 2]",
        "Explanation": "[Summarize why this answer is the
        overall winner based on the three criteria]"
    }
}
```

## D    PROOF OF PROPOSITION 1

**Proposition 1.** Given a graph-based RAG and a vanilla RAG system that share the same augmentation and generation processes, the absence of entity resolution causes the graph-based RAG to degrade into vanilla RAG.

*Proof.* We assume that: (1) both systems use identical augmentation and generation processes except for the knowledge representation, (2) vanilla RAG retrieves chunks based on relevance scoring, and (3) graph-based RAG retrieves subgraphs or triples based on query-entity matching. This is not a formal proof but rather an intuitive argument.

Given document chunks $\mathcal{C} = \{c_1, \ldots, c_M\}$, a Graph-based RAG system constructs a knowledge graph $\mathcal{G}^* = (\mathcal{E}^*, \mathcal{R}^*, \mathcal{T}^*, \mathcal{A}^*)$ through named entity recognition followed by deduplication. The

response $\mathcal{Y}$ is generated for query $Q$ as:

$$\mathcal{Y} = \mathcal{M} \circ \text{Aug}\big[Q, \text{Ret}(Q, \mathcal{G}^*)\big]. \tag{12}$$

Without entity resolution, the deduplication function becomes the identity mapping $\phi(e) = e$ for all $e \in \mathcal{E}_{\text{raw}}$. This means:

$$\mathcal{E}^* = \{\phi(e) \mid e \in \mathcal{E}_{\text{raw}}\} = \mathcal{E}_{\text{raw}} \tag{13}$$

$$\mathcal{T}^* = \mathcal{T}_{\text{raw}} \tag{14}$$

$$\mathcal{A}^*(e) = \mathcal{A}_{\text{raw}}(e) \quad \forall e \in \mathcal{E}^* \tag{15}$$

Since each triple $(e_1, r, e_2) \in \mathcal{T}_{\text{raw}}$ originates from a single chunk $c_m$, and no entity merging occurs, entities from different chunks remain disconnected even if they represent the same real-world concept. Formally, let $\mathcal{E}_m = \{e_1, e_2 \mid (e_1, r, e_2) \in \mathcal{T}_m\}$ be entities extracted from chunk $c_m$. Without entity resolution, there are no edges connecting entities from different chunks:

$$\forall i \neq j : \quad \mathcal{N}(e_i) \cap \mathcal{E}_j = \emptyset \quad \text{where } e_i \in \mathcal{E}_i \tag{16}$$

This results in $M$ disconnected subgraphs $\mathcal{G}_1^*, \mathcal{G}_2^*, \ldots, \mathcal{G}_M^*$, where each $\mathcal{G}_m^* = (\mathcal{E}_m, \mathcal{R}_m, \mathcal{T}_m, \mathcal{A}_m)$ corresponds to chunk $c_m$.

For any query $Q$, the graph retrieval function $\text{Ret}(Q, \mathcal{G}^*)$ can only retrieve from individual disconnected components. Since each component $\mathcal{G}_m^*$ contains only local information from chunk $c_m$, the retrieved content consists of triples $\mathcal{T}_m$ that represent structured partitions of the original chunk content. The graph-based retrieval without entity resolution becomes:

$$\text{Ret}(Q, \mathcal{G}^*) = \bigcup_{m:\text{rel}(Q, \mathcal{G}_m^*) > \tau} \mathcal{T}_m \tag{17}$$

where $\text{rel}(Q, \mathcal{G}_m^*)$ measures relevance between query and local subgraph, and $\tau$ is a threshold.

Note that each original chunk $c_m$ can be decomposed as:

$$c_m = \mathcal{T}_m \cup \text{unextracted text} \tag{18}$$

where $\mathcal{T}_m$ represents the structured information extracted from $c_m$. Since $\mathcal{T}_m \subset c_m$, the retrieved triples are essentially parts of the original chunks. With no cross-chunk connections, this retrieval process can be considered as a vanilla RAG system:

$$\text{Ret}_{\text{vanilla}}(Q, \{\mathcal{T}_m\}) = \{\mathcal{T}_m \mid \text{rel}(Q, \mathcal{T}_m) > \tau'\} \tag{19}$$

for appropriately chosen thresholds $\tau$ and $\tau'$.

Since the augmentation and generation processes are identical by assumption, and the retrieved content has the same information coverage (parts of chunks vs. disconnected subgraphs), we have:

$$\mathcal{Y}_{\text{graph}} = \mathcal{M} \circ \text{Aug}[Q, \text{Ret}(Q, \mathcal{G}^*)] \equiv \mathcal{M} \circ \text{Aug}[Q, \text{Ret}_{\text{vanilla}}(Q, \{\mathcal{T}_m\})] = \mathcal{Y}_{\text{vanilla}} \tag{20}$$

Therefore, without entity resolution, graph-based RAG degrades to vanilla RAG.

