# OpenReview forum: "Less is More: Denoising Knowledge Graphs For Retrieval Augmented Generation"
_ICLR.cc/2026/Conference — Submitted to ICLR 2026_

### Official Review · Reviewer_nAoT · 2025-10-29

**Soundness:** 3
**Presentation:** 3
**Contribution:** 2
**Rating:** 4
**Confidence:** 2

**Summary:**

The paper studies the effect of denoising LLM-generated knowledge graphs used in graph-based RAG and provides comprehensive analysis on the importance of entity resolution for graph RAG methods.
According to the paper, the proposed pipeline, DEG-RAG, consists of two core components:
•	Entity resolution: blocking → matching (using embeddings / neighbor info / types) → merging or synonym linking to collapse duplicates and near-duplicates; and
•	Triple reflection: an LLM “judge” scores triples and low-score triples are removed.
The authors evaluate multiple entity resolution design choices (blocking types, embedding families, similarity modes, merge strategies) across four Graph-RAG systems (LightRAG, HippoRAG, LGraphRAG, GGraphRAG) and four datasets from the UltraDomain benchmark (Agriculture, CS, Legal, Mix). Primary empirical claims: removing ≈40% of nodes + relations yields consistent QA improvements (winning-rate >50% in many settings), type-aware blocking tends to work best, classical KG embeddings can rival LLM embeddings, and direct merging often beats synonym-linking only. They also show robustness to aggressive reduction ratios (even ~70% in some datasets).

**Strengths:**

- This paper researches an important bottleneck in graph-based RAG pipelines—noise in LLM-generated KGs and empirically highlight the importance of entity resolution for graph RAG methods, i.e., without entity resolution, graph-based RAG will degrade into vanilla RAG.
- It ablates multiple entity resolution design component choices across several RAG systems and domains, and provides useful insights on the impacts of different entity resolution components on graph-based RAG systems, e.g., type-based blocking is more effective than semantic-based or structure-based blocking and classical KG embeddings can outperform LLM embeddings in efficiency and accuracy.
- The experimental results show that denoising improves QA despite heavy graph reduction and verifies the “less is more’’ effect.

**Weaknesses:**

- This paper does not introduce novel technical contribution but instead to improve the mined subgraph with existing entity resolution techniques.
- Lack of quantitative measurement of entity resolution quality. The paper emphasizes that the entity resolution step removes duplicates and merges entities of same identities. However: There is no ground-truth or gold dataset used to measure precision, recall, or F1 of merges vs. non-merges. As a result, it’s unclear whether the denoised graph is semantically cleaner or merely smaller. A single case study is provided but a systematic evaluation is missing.
- The evaluation heavily depends on LLM-based judgment rather than objective correctness, the  “win rate” between responses generated with/without denoising—is inherently subjective and model-dependent. This introduces potential concerns: the same biases or stylistic preferences may favor DEG-RAG outputs without reflecting actual factual correctness or reasoning quality. While this approach is becoming common in RAG evaluations, the absence of human annotation or reference-based metrics (e.g., factual F1, answer string match) makes it unclear whether improvements stem from better reasoning.

**Questions:**

Q1: I did not find a clear explanation about the entity reduction ratio, why this can be controlled, and how it is controlled. I guess this is done by randomly unmerging some of the merged nodes, but then the question is whether the result is sensitive to the selection of the unmerged nodes?
Q2: Is there any reason for setting the triple reflection threshold to 0.2? According to the scoring guidelines in Appendix C.4, “0.0–0.3: Invalid or highly questionable (factually wrong, illogical, meaningless)”, 0.2 does not look like a persuasive threshold.

---

### Official Review · Reviewer_14qP · 2025-10-30

**Soundness:** 3
**Presentation:** 3
**Contribution:** 2
**Rating:** 2
**Confidence:** 4

**Summary:**

The paper targets noise in LLM-constructed knowledge graphs used for Graph-based RAG. It proposes DEG-RAG, a two-step denoising pipeline: entity resolution to merge duplicate entities and triple reflection to filter incorrect relations. The authors run a systematic study of blocking, embeddings, similarity metrics, and merging strategies tailored to LLM-generated KGs. Experiments show the approach can remove about 40% of entities and relations while improving QA performance across several Graph-RAG variants, arguing that graph quality matters more than graph size for effective retrieval and generation.

**Strengths:**

- Clear problem focus: it tackles noise in LLM-built knowledge graphs, a practical bottleneck for Graph-RAG.

- Simple, modular pipeline: entity resolution plus triple filtering is easy to plug into existing Graph-RAG systems.

- Consistent gains: denoising improves QA quality across multiple Graph-RAG variants.

**Weaknesses:**

- Limited novelty.
The pipeline (entity resolution and triple filtering) is standard in knowledge graph cleaning [1,2]. The approach heavily relies on an LLM acting as the judge, rather than introducing a fundamentally new algorithm.

- Missing denoising baselines.
The paper does not compare against strong baselines on knowledge graph refinement, so the advantage over dedicated methods is unclear.

- Evaluation focuses on downstream QA only.
It rarely reports denoising metrics such as precision, recall, over-merge rate, or error rate.

- Transferability is uncertain.
Experiments use LLM-extracted graphs. It is unclear how well the method works on curated graphs like DBpedia or Wikidata, or on noisy industrial data.

- Scalability and maintenance are underexplored.
Blocking, embedding, similarity checks, merging, and re-indexing can be expensive on very large graphs. The cost is not well detailed.

- Risk of information loss.
Denoising may remove rare but important facts or collapse distinct entities.

[1] Knowledge graph refinement: A survey of approaches and evaluation methods[J]. Semantic web, 2016, 8(3): 489-508.

[2] Subagdja B, Shanthoshigaa D, Wang Z, et al. Machine learning for refining knowledge graphs: A survey[J]. ACM Computing Surveys, 2024, 56(6): 1-38.

**Questions:**

Transferability is uncertain. Experiments use LLM-extracted graphs. It is unclear how well the method works on curated graphs like DBpedia or Wikidata, or on noisy industrial data.

---

### Official Review · Reviewer_Nyok · 2025-10-31

**Soundness:** 3
**Presentation:** 3
**Contribution:** 2
**Rating:** 4
**Confidence:** 3

**Summary:**

This paper tackles the problem of noise and redundancy in LLM-generated knowledge graphs (KGs) used by graph-based retrieval-augmented generation (RAG) systems. The authors observe that current graph-RAG pipelines often suffer from duplicated entities (aliases, abbreviations, typos) and unreliable triples that degrade retrieval quality and increase computational overhead.
To address this, they propose DEG-RAG, a lightweight post-processing pipeline that denoises KGs through (1) entity resolution (ER)—which merges duplicate entities using blocking, embedding-based similarity, and merging/linking—and (2) triple reflection, which filters low-confidence relations based on an LLM-judged reliability score.
The cleaned graph is then reused by various existing graph-RAG retrievers (LightRAG, HippoRAG, LGraphRAG, GGraphRAG). Empirical results on the UltraDomain benchmark show consistent improvements in LLM-judged answer quality (≈5–15% win-rate gains) while reducing the KG size by ~40%. The authors also perform ablations on each design choice (blocking strategy, embedding type, merging method, threshold) and show robustness to high pruning ratios and generality across different LLM backbones.

**Strengths:**

Clear and impactful motivation — The issue of noisy LLM-generated KGs is real and under-explored; the authors address it with a practical, model-agnostic approach.

Simple but effective methodology — The proposed ER + triple-filtering pipeline is lightweight, interpretable, and compatible with existing graph-RAG frameworks.

Comprehensive experiments — Multiple graph-RAG baselines, four domains, and detailed ablations (blocking, embeddings, thresholds) make the evaluation convincing.

Strong empirical gains with efficiency — DEG-RAG improves answer quality while reducing graph size, supporting the “less is more” thesis.

Insightful analysis — The observation that graph-RAG degenerates to chunk-RAG without entity resolution is novel and theoretically motivated.

**Weaknesses:**

Limited theoretical grounding — While intuitively reasonable, the paper lacks a formal analysis of why denoising improves reasoning (e.g., retrieval connectivity, semantic precision).

Dependence on LLM scoring — Triple reflection uses an LLM-as-judge, which may introduce bias or high variance; the reliability of this signal is not deeply validated.

Evaluation via LLM-as-judge only — No human or retrieval-based metrics are reported; LLM-judged win-rates might not fully capture factual correctness.

Domain generalization — Experiments are limited to four UltraDomain subsets; it is unclear whether the approach scales to open-domain or cross-lingual corpora.

**Questions:**

Complexity and scalability: What is the computational complexity of the ER and triple reflection steps? How does the runtime scale with the number of entities/triples?

Sensitivity to parameters: How sensitive are the results to the ER threshold (δ_ER) and triple score threshold (δ_TR)? Are there guidelines for tuning them automatically?

Triple reflection reliability: How consistent are the LLM-as-judge scores across different models (e.g., Qwen vs GPT-4o)? Have you measured inter-model agreement?

Graph structure preservation: Does merging entities or removing triples ever break necessary connectivity (e.g., for multi-hop reasoning)? How is information loss mitigated?

Downstream generalization: Have you tried DEG-RAG on non-UltraDomain datasets or on real-world document corpora (e.g., academic papers, Wikipedia)?

---

### Official Review · Reviewer_yvcH · 2025-11-02

**Soundness:** 2
**Presentation:** 2
**Contribution:** 2
**Rating:** 2
**Confidence:** 5

**Summary:**

The paper proposes a method to denoise LLM-generated knowledge graphs for Graph RAG, using entity resolution and a triple-reflection step to remove redundant entities and filter unreliable relations.

**Strengths:**

1. Tackles a timely, important problem for both industry and academia.
2. Combines entity resolution with triple reflection to prune redundant nodes and questionable relations.
3. Empirically robust performance that remains stable even after removing up to ~70% of entities.

**Weaknesses:**

1.	Key implementation details are omitted, e.g., how canonical entities are chosen and how retrieval operates after merging.
2.	Triple reflection relies on LLM-as-judge, which incurs high computational cost and may be impractical at scale.
3.	Evaluation depends on LLM-as-judge metrics, which are sensitive to model choice and prompts and may not reliably reflect true method quality.

**Questions:**

1.	For entity resolution, how are canonical entities selected and evaluated for representativeness? After merging, both original and merged entities appear in the KG. How does retrieval handle this, and how are hyperparameters (e.g., top-k) chosen?
2.	For triple reflection, since each triple is judged by an LLM, runtime and cost could be prohibitive for large KGs. Can the authors propose or test lightweight alternatives, e.g., learned classifiers, heuristics, or ensemble-based filters, that approximate LLM judgments?
3.	For performance evaluation, given the sensitivity and inconsistency of LLM-as-judge, please include alternative or complementary metrics for KG quality to better validate the denoising effectiveness across methods.

---

### Meta-Review · Area_Chair_bvKi · 2026-01-08

**Summary:**

The paper addresses the issue of noise and redundancy in knowledge graphs automatically generated by LLMs for Graph-based RAG. The authors propose DEG-RAG, a framework comprising entity resolution to consolidate duplicate entities, and triple reflection (using an LLM-as-judge) to filter unreliable relations. The key insight is that applying these denoising steps reduces graph size by approximately 40% while consistently improving downstream QA performance across multiple Graph-RAG architectures.

**Reviewer Concerns:**

The authors did not provide a rebuttal or response to the reviewer comments; therefore, all concerns raised by the reviewers remain unaddressed.

Outstanding Concerns:
1. Lack of Intrinsic Evaluation: Reviewers 14qP and nAoT highlighted that the paper does not report standard denoising metrics (e.g., Precision, Recall, F1) using a ground-truth dataset. Consequently, it is unclear whether the method accurately resolves entities or simply aggressively merges them.

2. Limited Novelty: Reviewers 14qP and nAoT noted that the proposed pipeline relies on standard, existing Entity Resolution techniques and simple LLM prompting. The contribution is seen as an application of established methods rather than a novel algorithmic advancement.

3. Reliance on LLM-as-Judge: Reviewers yvcH and Nyok expressed strong concerns regarding the exclusive reliance on LLM-based metrics for evaluation, citing potential biases, high variance, and the lack of correlation with human judgments or objective correctness.

4. Implementation & Cost: Reviewers yvcH and Nyok questioned the scalability of the Triple Reflection step (which requires an LLM call per triple) and noted missing details regarding canonical entity selection and hyperparameter sensitivity.

**Reviewer Scores:**

no changes

---

### Decision · Program_Chairs · 2026-01-26

Reject